# Tat-Beclin-1 Peptide Ameliorates Metabolic Dysfunction-Associated Steatotic Liver Disease by Enhancing Hepatic Autophagy

**DOI:** 10.3390/ijms252212372

**Published:** 2024-11-18

**Authors:** Chun-Liang Chen, Fen-Fen Huang, Hsueh-Fang Lin, Chi-Chien Wu, Yen-Hsuan Ni, Yu-Cheng Lin

**Affiliations:** 1Department of Pediatrics, Taipei Veterans General Hospital, Taipei 112, Taiwan; muzhikmuzhik@gmail.com (C.-L.C.); d6169kimo@gmail.com (H.-F.L.); 2Department of Healthcare Administration, Asia Eastern University of Science and Technology, New Taipei City 220, Taiwan; fl005@mail.aeust.edu.tw; 3Department of Pediatrics, Far Eastern Memorial Hospital, New Taipei City 220, Taiwan; kevin200074@gmail.com; 4Department of Pediatrics, National Taiwan University Hospital, Taipei 100, Taiwan; yhni@ntu.edu.tw

**Keywords:** MASLD, autophagy, steatosis, fibrosis, lipid metabolism

## Abstract

Autophagy plays a crucial role in hepatic lipid metabolism, making it a key therapeutic target for addressing metabolic dysfunction-associated steatotic liver disease (MASLD). This study evaluates the efficacy of the Tat-Beclin-1 (TB-1) peptide, a specific autophagy inducer, in mitigating MASLD. Initially, we examined the impact of the TB-1 peptide on autophagic activity and intracellular lipid metabolism in HepG2 cells treated with oleic acid, using a Tat scrambled (TS) control peptide for comparison. Subsequently, we established a MASLD mouse model by feeding a high-fat diet (HFD) for 16 weeks, followed by intraperitoneal administration of TB-1 or TS. Assessments included liver histopathology, serum biochemistry, and autophagy marker analysis. Our findings indicate that the TB-1 peptide significantly increased the LC3II/β-actin ratio in a dose- and time-dependent manner while promoting the expression of key autophagy markers Beclin-1 and ATG5-12. Furthermore, TB-1 treatment led to a marked reduction in both the size and number of lipid droplets in HepG2 cells. In vivo, HFD-fed mice exhibited increased liver weight, elevated serum alanine aminotransferase levels, and impaired oral glucose tolerance. TB-1 administration effectively mitigated these hepatic and metabolic disturbances. Histological analysis further revealed a substantial reduction in the severity of hepatic steatosis and fibrosis in TB-1-treated mice compared to TS controls. In conclusion, the TB-1 peptide shows significant potential in reducing the severity of MASLD in both HepG2 cell models and HFD-induced MASLD mouse models. Enhancing autophagy through TB-1 represents a promising therapeutic strategy for treating MASLD.

## 1. Introduction

The development and progression of metabolic dysfunction-associated steatotic liver disease (MASLD) involves multiple factors and is strongly associated with obesity and insulin resistance [1]. However, the pathogenesis of MASLD is complex and remains largely elusive, making drug development challenging [2]. Autophagy, an essential lysosomal degradation pathway in maintaining energy homeostasis and protein/organelle quality control, has been recognized as a crucial player in hepatocyte lipid metabolism [3,4]. Evidence suggests that decreased autophagy promotes the development of hepatic steatosis and the progression of liver damage. Modulating hepatic lipid metabolism through autophagy regulation may hold promise as a therapeutic target for MASLD [5].

Tat-Beclin-1 (TB-1) peptide, derived from the Beclin-1 protein, a key regulator of autophagy, and linked to the HIV-1 Tat protein, has shown potential for specifically inducing autophagy. TB-1 peptide selectively induces autophagy by interacting with the negative regulator of autophagy, Golgi-Associated Plant Pathogenesis-Related protein 1 (GAPR-1), without non-specifically affecting other pathways [6].

Although various autophagy inducers have been investigated for treating MASLD, TB-1 peptide can more directly induce autophagy processes. This specificity may reduce unnecessary systemic side effects, thereby enhancing treatment safety. Furthermore, existing studies have shown significant therapeutic effects of TB-1 peptide in models of Parkinson’s disease and sepsis without showing obvious toxicity [6,7]. These findings suggest that Tat-Beclin 1 has good biocompatibility and therapeutic potential.

In contrast, other autophagy inducers like rapamycin and metformin, while showing potential benefits for MASLD in some studies, have broader mechanisms of action that might affect multiple cellular processes, leading to adverse side effects. For example, rapamycin, an mTOR inhibitor, not only activates autophagy but also inhibits protein synthesis and cell proliferation, leading to many side effects [8]. Metformin activates autophagy indirectly through AMPK activation, but AMPK is also involved in regulating other metabolic processes [9]. Minimizing potential off-target effects is crucial for the development and implementation of autophagy-based therapies for MASLD.

In this study, we aim to investigate the potential of TB-1 peptide, a potent and specific autophagy inducer, in improving MASLD using a high-fat diet (HFD)-induced murine model.

## 2. Results

### 2.1. Autophagy Markers After TB-1 Peptide Treatment in HepG2 Cells

To investigate the effects of TB-1 peptide on liver autophagy, we assessed its impact on autophagy marker protein expression in HepG2 cells. To measure autophagic flux, cells were treated with the autophagy inhibitor bafilomycin A1 for 2 h before harvesting for Western blot analysis [10,11].

Following a 24 h treatment with 10, 30, and 50 μM TB-1 peptide, there was a significant induction of autophagy markers LC3-II, ATG5-12, and Beclin-1, while p62 expression was notably reduced (Figure 1A,B). Furthermore, treatment with 30 μM TB-1 peptide for 2, 4, 8, 16, and 24 h in HepG2 cells resulted in a time-dependent increase in LC3-II, ATG5-12, and Beclin-1 levels, accompanied by a gradual decrease in p62 expression (Figure 1C,D). These findings demonstrate that TB-1 peptide, but not the TS control peptide, induces autophagy in a dose- and time-dependent manner.

Autophagic vacuole staining using CYTO-ID^®^ dye confirmed these results, showing an increase in the number and size of autophagic vacuoles in HepG2 cells treated with TB-1 peptide (Appendix A).

### 2.2. Reduced Lipid Droplet Content in HepG2 Cells After TB-1 Peptide Treatment

Next, we investigated the impact of TB-1 peptide on lipid content in the human hepatocyte cell line. HepG2 cells were pre-treated with 0.1 mM oleic acid overnight to increase the baseline lipid content. Subsequently, the cells were treated with 10 μM, 30 μM, and 50 μM TB-1 peptide for 24 h (Figure 2A).

Compared to the TS control peptide, the average number of lipid droplets (LD) per cell and LD size in HepG2 cells treated with 10 μM TB-1 peptide decreased by 32.6% and 18.6%, respectively (Figure 2B). Furthermore, HepG2 cells treated with 30 μM or 50 μM TB-1 peptide showed an even greater reduction in LD number by up to 60% and in LD size by up to 35% compared to the TS control peptide (Figure 2B). In contrast, the 50 μM TS control peptide did not show any reduction in intracellular lipid droplet content in HepG2 cells (Figure 2A,B).

### 2.3. TB-1 Peptide Induced Autophagic Vacuoles Formation in Human Hepatocyte

We next investigated the effects of TB-1 peptide on autophagic vacuole formation. To measure autophagic flux, we treated the cells with the autophagy inhibitor, bafilomycin A1, which inhibits the fusion of autophagosomes and lysosomes, 2 h before staining the autophagic vacuoles. The average number of autophagic vacuoles per HepG2 cell was 1.693 at baseline, and increased to 5.576 after treatment with 30 μM TB-1 peptide (Appendix A). The average size of autophagic vacuoles was 1.748 pixels^2^ at baseline, and increased to 2.117 pixels^2^ after treatment with 30 μM TB-1 peptide (Appendix A).

### 2.4. Effects of TB-1 Peptide on Body Weight in HFD-Induced Obese Mice

To investigate the effects of TB-1 peptide in vivo, we treated HFD-fed mice with TB-1 peptide (HFD + TB-1) or TS control peptide (HFD + TS) for 3 weeks and compared them with chow diet (CD)-fed mice treated with TS control peptide (CD + TS). Health status and weight were monitored weekly. HFD groups experienced greater weight gain than the CD + TS control group. However, no significant difference in body weight was observed between the HFD + TB-1 group and the HFD + TS group (Appendix A). The mice displayed no signs of peritonitis, such as reduced appetite, decreased activity, kyphosis, abdominal fluid accumulation, or diarrhea. These findings suggest that the effect of TB-1 peptide on attenuating hepatic steatosis and fibrosis is independent of body weight changes or peritonitis resulting from repeated intraperitoneal injections.

### 2.5. Effects of TB-1 Peptide on Blood Biochemical Parameters in HFD-Induced Obese Mice

The effects of TB-1 peptide on blood biochemical parameters are shown in Table 1. HFD feeding (HFD + TS and HFD + TB-1) resulted in significant increases in serum levels of AST, ALT, and T-CHO compared to the CD + TS group. However, treatment with TB-1 peptide significantly improved the elevated levels of AST and ALT compared to the HFD + TS group.

To assess the effects of TB-1 peptide on blood glucose levels, an OGTT assay was performed (Appendix A). HFD feeding resulted in a significant increase in the area under the curve (AUC) of OGTT (Appendix A). TB-1 peptide treatment significantly improved glucose intolerance induced by the high-fat diet.

### 2.6. Improved Hepatic Steatosis and Fibrosis After TB-1 Peptide Treatment in HFD-Induced Obese Mice

To assess hepatic steatosis severity, we performed H and E staining on liver tissue sections. HFD feeding significantly increased hepatic steatosis to 40%. However, TB-1 peptide administration reduced this to 20.3% (Figure 3A,C). These results suggest that TB-1 peptide has a beneficial effect on reducing lipid accumulation in the liver.

For liver fibrosis evaluation using Masson’s trichrome staining, the HFD-induced murine MASLD model showed a liver fibrosis level of 7.3%. Treatment with TB-1 peptide significantly reduced this to 3.1% (Figure 3B,D), suggesting that TB-1 peptide can attenuate liver fibrosis in MASLD.

### 2.7. TB-1 Peptide Strongly Activates Liver Autophagy in HFD-Induced Obese Mice

To investigate the impact of TB-1 peptide on liver autophagy in an HFD-induced murine MASLD model, we assessed the expression levels of key autophagy markers through Western blot analysis (Figure 4A,B).

HFD feeding significantly increased the levels of autophagy markers, including LC3-II, Beclin-1, SIRT1, and pULK1/tULK1, while significantly reducing the expression of p62, a protein degraded during autophagy. This indicates an upregulation of autophagy in response to HFD-induced MASLD.

TB-1 peptide treatment further augmented autophagy activity, with significantly increased levels of LC3-II, Beclin-1, SIRT1, and pULK1/tULK1, and a more pronounced decrease in p62 expression. These results suggest that TB-1 peptide greatly enhances liver autophagy in the context of HFD-induced MASLD.

### 2.8. Upregulated Fatty Acid Oxidation-Related Gene Expression After TB-1 Peptide Treatment in HFD-Induced Obese Mice

Next, to investigate the effects of TB-1 peptide on lipid metabolism in the liver, we examined the mRNA expression levels of key fatty acid oxidation-related genes, including *Ucp2*, *Cpt-1α*, *Lcad*, *Acox*, and *Ppar-α* (Figure 5A).

HFD feeding increased the mRNA expression levels of *Ucp2* and *Cpt-1α* in the HFD + TS group compared to the CD + TS group, while *Lcad* and *Ppar-α* expression levels were slightly decreased. However, TB-1 peptide treatment significantly increased the expression of all these fatty acid oxidation-related genes. Notably, *Ucp2* and *Cpt-1α* expression levels dramatically increased by 7.6-fold and 9.8-fold, respectively, compared to the HFD + TS group.

These findings indicate that TB-1 peptide enhances the expression of fatty acid oxidation-related genes in the context of HFD-induced MASLD, suggesting a potential mechanism for its beneficial effects on lipid metabolism.

### 2.9. TB-1 Peptide Enhances Mitochondrial Fatty Acid Oxidation

Fatty acid oxidation was assessed using the SeaHorse Flux Analyzer with the XF Long Chain Fatty Acid Oxidation Stress Test Kit. Following 16 h incubation, 10 μM TB-1 peptide treatments resulted in significant increases in mitochondrial basal respiration by 1.96-fold, compared to the TS control peptide group (Figure 5B,C). Furthermore, mitochondrial maximal respiration in HepG2 cells increased by 1.88-fold in response to 10 μM TB-1 peptide treatments, respectively (Figure 5B,D). Additionally, ATP production activity increased by 1.93-fold with 10 μM TB-1 peptide compared to the TS control peptide group in HepG2 cells (Figure 5B,E). These findings suggest that TB-1 peptide activates autophagy and promotes lipid degradation to fuel mitochondrial oxidative phosphorylation.

## 3. Discussion

Our study investigated the effectiveness of TB-1 peptide, an autophagy inducer, in alleviating MASLD severity. In HepG2 cells, TB-1 peptide induced autophagic vacuoles, increased autophagy-related gene expression, and decreased lipid droplets. In an MASLD mouse model, TB-1 treatment ameliorated hepatic and metabolic abnormalities, resulting in reduced hepatic steatosis and fibrosis. These findings suggest that TB-1 peptide is a potential therapeutic agent for MASLD.

Autophagy plays a crucial role in maintaining physiological balance, both in normal functioning and in disease states. It is associated with conditions like cancer, neurodegeneration, and cardiac disorders, making it an attractive target for therapeutics [12]. Despite significant efforts, no autophagy-selective drugs have been developed for clinical use. Existing drugs like metformin affect a wide range of molecules and pathways, lacking specificity for the autophagy pathway. TB-1 is a synthetic peptide designed to mimic Beclin 1, a key autophagy regulator [6]. This targeted specificity may reduce unnecessary side effects, thereby enhancing the safety and efficacy of treatments.

Several studies have highlighted the unique properties of the TB-1 peptide. In cognitive impairment models, the TB-1 peptide improved memory by promoting the autophagy-mediated degradation of amyloid fibrils [13]. It also showed protective effects in infection models of Sindbis virus, chikungunya virus, West Nile virus, and HIV-1 by clearing protein aggregates [6]. In a rodent sepsis model, the TB-1 peptide improved cardiac function, reduced inflammation, and rescued phenotypes caused by *Beclin 1* deficiency [7]. In our study, we similarly demonstrated that the TB-1 peptide reduced liver inflammation in MASLD, as indicated by decreased expression of *Il-1β* and *Tnf-α* genes in liver tissue (Appendix A). Additionally, the TB-1 peptide reduced ocular hypertension in a murine glaucoma model through autophagic degradation of mutant myocilin [14]. The TB-1 peptide also rescued retinal neurons from cell death after oxygen-glucose deprivation [15]. Furthermore, the TB-1 peptide inhibited the tumorigenesis of dual specificity phosphatase 4-positive papillary thyroid carcinoma cells [16]. These findings demonstrate the broad therapeutic potential of TB-1 in various diseases, from neurodegenerative conditions to infections and ocular disorders.

Research on TB-1 in treating liver diseases is limited. In an in vitro cell study, polylactic acid polymer-coated TB-1 peptide induced autophagy and reduced intracellular lipid droplets in a cell model [17]. Our study builds on this by providing the first in-depth in vivo evidence of TB-1’s therapeutic effect on MASLD, demonstrating significant benefits without obvious side effects at the doses used. Additionally, the TB-1 peptide has been explored in other liver-related diseases. Soria et al. showed that TB-1 peptide activated liver autophagy, increased ureagenesis, and protected against hyperammonemia in both acute and chronic hyperammonemia animal models [18].

In conclusion, our study demonstrated that enhancing autophagy with the TB-1 peptide could be a beneficial strategy for managing MASLD by improving lipid metabolism, reducing hepatic steatosis, and alleviating liver fibrosis (Figure 6). Continued research and clinical trials will further elucidate its role and potential in the treatment landscape for MASLD.

## 4. Materials and Methods

### 4.1. Cell Culture

HepG2 (BCRC No. 60177, Hsinchu, Taiwan) cells were obtained from Bioresource Collection and Research Center (BCRC, Taiwan) and cultured in Modified Eagle’s Medium (MEM, Gibco™, 11095080, Vilnius, Lithuania) supplemented with 10% fetal bovine serum (FBS) and 100 U/mL Penicillin/Streptomycin (Thermo Fisher Scientific, 15140122, Vilnius, Lithuania). Cells were incubated in an incubator with 5% CO_2_ at 37 °C. To induce lipid droplet accumulation, the complete cell growth medium was supplemented with 0.1 mM oleic acid-albumin (Sigma-Aldrich, O3008, Darmstadt, Germany) and incubated for 16 h.

### 4.2. Lipid Droplet Quantification by Oil Red O Staining

The cells were fixed in 4% paraformaldehyde and stained with Oil Red O (Sigma-Aldrich, O1391, Darmstadt, Germany). Lipid droplets were visualized using a bright-field microscope, and their quantification was performed using Image J software (version 1.53k). Red pixels were identified by analyzing the color spectra. The number of lipid droplets was quantified using the “analyze particles” function in Image J. A minimum of 20 cells per slide were analyzed, and the settings for particle size (pixel^2^) were adjusted from 0.1 to 10, with circularity from 0 to 1.

### 4.3. Autophagic Vacuole Staining

After 4 h of treatment with Tat-Beclin-1 peptide (TB-1, Sigma-Aldrich, 5.06048, Darmstadt, Germany) or Tat-scrambled control peptide (TS, Sigma-Aldrich, 5.31038, Darmstadt, Germany), HepG2 cells were treated with 400 nM Bafilomycin A1 (Sigma-Aldrich, 88899-55-2, Darmstadt, Germany) for 2 h before staining. Autophagic vacuoles were stained by CYTO-ID^®^ Autophagy detection kit 2.0 (Enzo Life Sciences, ENZ-KIT175, Farmingdale, NY, USA) following the manufacturer’s instructions. The resulting green fluorescence was visualized using a fluorescence microscope (ZEISS Axio Imager Z2, Oberkochen, Germany). The signals were quantified using Image J software (version 1.53k), and the values were analyzed.

### 4.4. Animals and TB-1 Peptide Treatment

Male C57BL/6JNarl mice (National Laboratory Animal Center, Taipei, Taiwan) were housed in cages at 22 °C under a 12:12 h light-dark cycle. After reaching sexual maturation (day 60), the mice were fed either a chow diet (CD, 10 kcal% fat, D12450B, Research Diets, Inc., New Brunswick, NJ, USA) or a high-fat diet (HFD, 60 kcal% fat + 1.25% cholesterol, customized D12492, Research Diets, Inc., New Brunswick, NJ, USA). The addition of cholesterol in the HFD contributed to more prominent hepatic fibrosis development within the 16-week feeding duration, making fibrosis changes more noticeable [19]. After 16 weeks on the HFD, the mice in the treatment groups were injected intraperitoneally (i.p.) with TB-1 peptide at a dose of 20 mg/kg body weight [20,21], while the mice in the control groups were treated with TS control peptide with the same route and dose. Both TS and TB-1 peptides are water-soluble, with a solubility of 50 mg/mL. Therefore, TS and TB-1 peptides were prepared in 1 × PBS for i.p. injection in the mice.

The animals were randomly divided into three groups: CD control (*n* = 10), HFD + TS (negative control peptide, *n* = 10), and HFD + TB-1 (*n* = 10) groups. After 16 weeks of HFD administration, the mice in the treatment group were injected i.p. with TB-1 peptide or TS three times per week for three weeks.

At the end of the experiment, mice were weighed and euthanized by carbon dioxide inhalation, blood was drawn by cardiac puncture, and the liver was excised, weighed, and apportioned for RNA/DNA extraction or liver histopathology as fresh-frozen tissue or preserved in 4% paraformaldehyde for fixation (PFA).

### 4.5. Liver Steatosis Assessment

For light microscopic analysis of liver histology, the paraffin-embedded liver tissues were sliced into thin sections, and standard hematoxylin and eosin (H and E) staining was performed. For light microscopic analysis of liver histology, paraffin-embedded liver tissues were sectioned thinly, and standard hematoxylin and eosin (H and E) staining was performed. The degree of steatosis was assessed by analyzing the H and E-stained sections. The percentage of LD area relative to the total area was quantified as a measure of steatosis severity.

### 4.6. Liver Fibrosis Quantification by Collagen Proportionate Area (CPA)

Hepatic fibrosis was assessed using Masson’s trichrome (TRI) staining, and the collagen proportionate area (CPA) was measured as previously described [22,23]. Images were captured using the Nuance FX multispectral imaging system (PerkinElmer, Waltham, MA, USA). After capturing the whole section’s digital image, CPA was measured using the inForm advanced image software 2.3 (PerkinElmer, Waltham, MA, USA). A threshold was set to detect areas of stained collagen based on the RGB (Red, Green, Blue) spectrum, and the collagen mask was calculated as an area in pixels. The CPA was expressed as a percentage. The CPA measurement included a manual editing step to eliminate image artifacts, and operator-dependent thresholding was used to determine the stained area of the section. To enhance consistency in our fibrosis measurements, this approach applied consistent criteria across all samples, which minimizes variability and supports uniformity in the fibrosis assessment process.

### 4.7. Blood Biochemical Parameters

Blood samples were collected in the 19th week. The serum levels of glucose (GLU), alanine aminotransferase (ALT), alanine transaminase (AST), total cholesterol (T-CHO), and triglycerides (TG) were measured by National Laboratory Animal Center (NLAC), Taiwan. The CD + TS group served as the baseline reference for comparing treatment responses.

### 4.8. Oral Glucose Tolerance Tests (OGTT)

After a 12 h fast, the mice were orally administered dextro-glucose (2 mg/g body weight) dissolved in PBS. Blood glucose levels were measured at 0, 15, 30, 45, 60, and 120 min after glucose administration using a Roche ACCU-CHEK^®^ Instant blood glucometer (Shanghai, China).

### 4.9. Cell and Tissue Lysate Preparation and Western Blot Analysis

Cells and crushed liver tissues were lysed in 100 µL TEGN lysis buffer [10 mM Tris base pH 7.5, 1 mM EDTA, 10% glycerol, 0.5% NP-40, 420 mM NaCl, and 1 × PhosSTOP™ (Roche, 4906845001, Darmstadt, Germany)] containing 1 × Complete™ Protease Inhibitor Cocktail (Roche, 11697498001, Darmstadt, Germany). After complete cell lysis confirmed by phase-contrast microscopy, the samples were centrifuged at 12,000× *g* for 30 min, and the supernatants were collected. Protein concentration was determined using the Pierce™ BCA Protein Assay Kit (Thermo Fisher Scientific, 23225, Vilnius, Lithuania) with bovine serum albumin as a standard.

Western blotting was performed using primary antibodies for LC3B (Cell Signaling, 3868, Danvers, MA, USA), SQSTM1/p62 (Abcam, ab109012, Cambridge, UK), Beclin-1 (Cell Signaling, 3495, Danvers, MA, USA), ATG5-12 (Cell signaling, 4180s, Danvers, MA, USA), SIRT1 (Proteintech, 13161-1-AP, Rosemont, IL, USA), ULK1 (Cell signaling, 8054, Danvers, MA, USA), and phospho-ULK1 (pSer317, Cell signaling, 12753, Danvers, MA, USA). These primary antibodies were diluted 1000-fold for use, except for SQSTM1/p62 (10,000-fold dilution) and GAPDH (20,000-fold dilution). A secondary antibody, goat anti-rabbit IgG H and L (HRP), was used to detect the primary antibodies. GAPDH was blotted using a Rabbit anti-GAPDH primary antibody as a protein loading control. Western blot results were repeated at least three times.

### 4.10. Mouse Liver RNA Extraction and Quantitative RT-PCR Analysis

Total RNA was extracted from fresh-frozen liver samples using the RNA Mini RNA isolation kit (GE Healthcare, 25-0500-71, Buckinghamshire, UK). cDNA was then synthesized using the High-Capacity cDNA Reverse Transcription Kit (ThermoFisher Scientific, 4368814, Vilnius, Lithuania). Quantitative RT-PCR was performed using the LightCycler^®^ 480 Real-Time PCR System (Roche v1.5) to measure the expression levels of genes related to fatty acid oxidation and inflammation (*Il-1β* and *Tnf-α* genes). The specific primers used in RT-PCR were summarized in Appendix A.

### 4.11. Fatty Acid Oxidation Measurement

The measurement of fatty acid oxidation was conducted using the XF Long Chain Fatty Acid Oxidation Stress Test Kit (Agilent, 103672-100, Cedar Creek, TX, USA) following the manufacturer’s instructions. HepG2 cells were first pre-cultured in SeaHorse 8-well plates with 0.1 mM OA for 6 h. Then, they were treated with either 50 μM TS control peptide or 10 μM TB-1 peptide for 16 h.

One hour before the assay, cells were washed and XF DMEM (pH 7.4, Agilent, 103575-100, Cedar Creek, TX, USA) was used, containing 10 mM glucose solution (Agilent, 103577-100, Cedar Creek, TX, USA), 1.0 mM pyruvate solution (Agilent, 103578-100, Cedar Creek, Texas, USA), and 2.0 mM glutamate solution (Agilent, 103579-100, Cedar Creek, TX, USA). The oxygen-consumption rate (OCR) of cells was analyzed using the SeaHorse Flux Analyzer (Seahorse XF HS Mini Analyzer, Agilent, S7852A, Cedar Creek, TX, USA) and normalized based on the total cellular protein concentration in each well. The levels of mitochondrial basal respiration, maximal respiration, and ATP production were calculated using Seahorse Wave Pro Software (version 10.1.0.1).

### 4.12. Statistical Analysis

Statistical analyses were performed using Stata statistical software (version 18, College Station, TX, USA). All results were presented as the mean ± S.D. Statistical analysis was performed using Student’s t-test for unpaired samples, and the significance level was set at a *p*-value of 0.05.

## Figures and Tables

**Figure 1 ijms-25-12372-f001:**
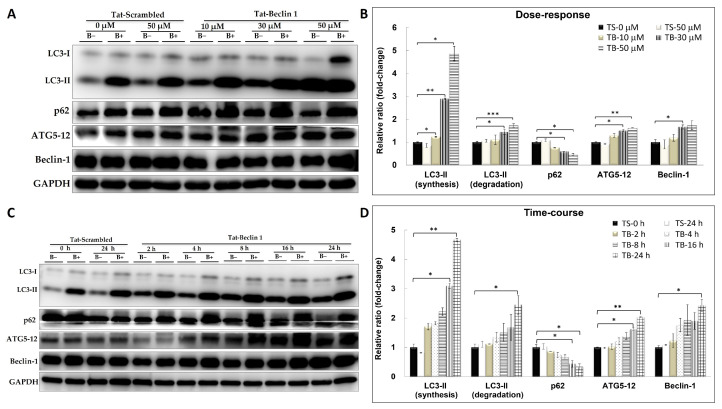
Induction of autophagy marker expression in HepG2 cells by TB-1 peptide. HepG2 cells were treated with TB-1 peptide or TS control peptide at various concentrations (**A**,**B**) and for different durations (**C**,**D**). B+ and B− represent conditions with or without bafilomycin A1 treatment, respectively. Immunoblot analyses were conducted to quantify the levels of autophagy marker proteins. The fold change for each target protein is shown relative to the TS control sample. Statistical significance: * *p* < 0.05, ** *p* < 0.01, and *** *p* < 0.001. TS: Tat-Beclin-1 scrambled control peptide; TB: Tat-Beclin-1 peptide.

**Figure 2 ijms-25-12372-f002:**
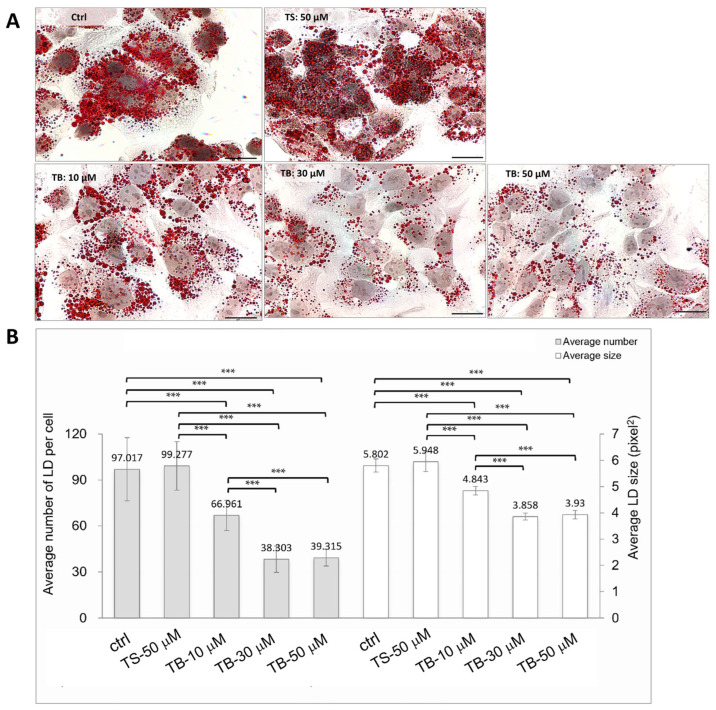
Reduction in intracellular lipid droplet content in HepG2 cells by TB-1 peptide. The upper panel shows Oil Red O staining of HepG2 cells treated with medium alone or TS control peptide. Treatment with TB-1 peptide at concentrations of 10, 30, and 50 μM resulted in a significant, dose-dependent decrease in lipid droplet content (red) (**A**). The lower panel presents the quantification of the average number and size of lipid droplets per cell (**B**). Lipid droplets were quantified using ImageJ software (version 1.53k), with at least 20 cells analyzed per experiment to determine the average number and size. Data are presented as the mean ± SD of three independent experiments. The scale bar represents 50 μm. Statistical significance: *** *p* < 0.001. TS: Tat-Beclin-1 scrambled control peptide; TB: Tat-Beclin-1 peptide.

**Figure 3 ijms-25-12372-f003:**
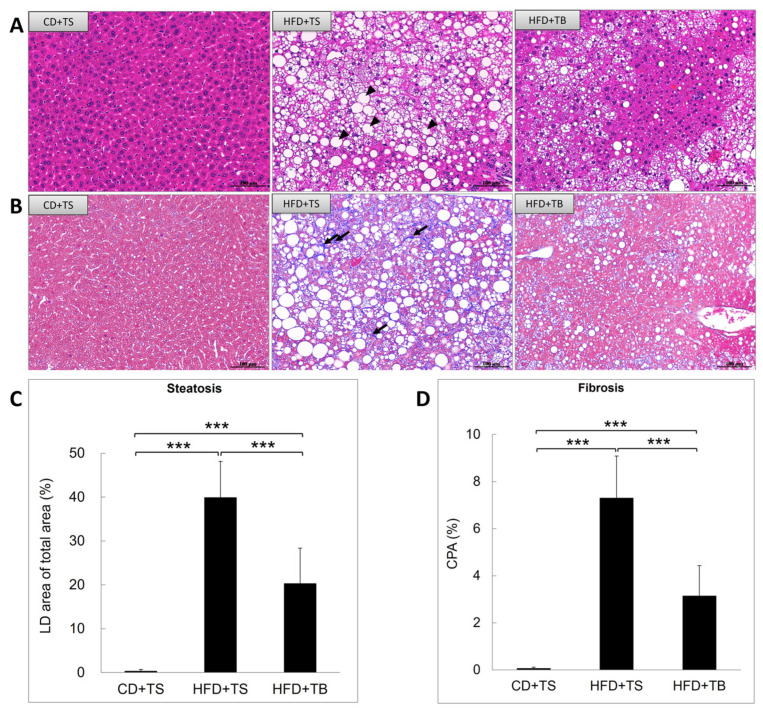
Reduction in liver steatosis and fibrosis in the HFD-induced murine MASLD model by TB-1 peptide. Liver histology was evaluated using H and E staining to assess liver steatosis, indicated by black arrow heads (**A**) and Masson’s trichrome staining to evaluate liver fibrosis, indicated by black arrows (**B**) following treatment with the TB-1 peptide. Liver steatosis (**C**) and fibrosis (**D**) were quantified in all samples using ImageJ software (version 1.53k) and collagen proportionate area (CPA) measurement. Data are presented as mean ± SD (*n* = 10 per group). The scale bar represents 100 μm, with a magnification of 200×. Statistical significance: *** *p* < 0.001. Abbreviations: HFD, high-fat diet; CD, chow diet; H and E, hematoxylin and eosin; CPA, collagen proportionate area; LD, lipid droplet; TS, Tat scrambled control peptide; TB, Tat-Beclin-1 peptide.

**Figure 4 ijms-25-12372-f004:**
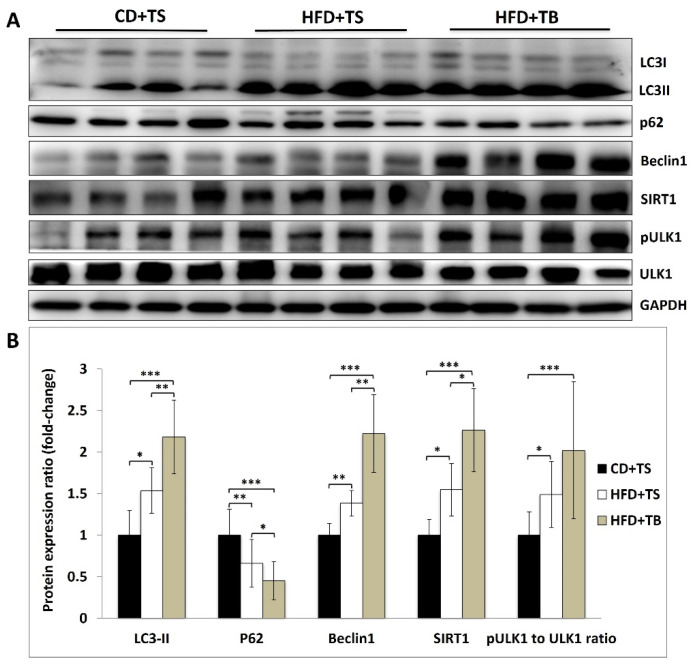
Induction of autophagy in the liver of HFD-induced murine MASLD model by TB-1 peptide. (**A**) Protein expression levels in the liver of mice were analyzed using Western blotting. (**B**) The immunoblots were quantified using Image J software (version 1.53k), with results presented as mean ± SD (*n* = 10 per group). The blots show the effects of TB-1 peptide on the levels of autophagy-related proteins, including LC3-II, SQSTM1/p62, Beclin-1, SIRT1, and the phospho-ULK1 (Ser 317) to total ULK1 ratio. Statistical significance: * *p* < 0.05, ** *p* < 0.01, and *** *p* < 0.001. Abbreviations: CD, chow diet; HFD, high fat diet; TS, Tat scrambled control peptide; TB, Tat-Beclin-1 peptide; SIRT1, Sirtuin 1; ULK1, Unc-51-like kinase 1.

**Figure 5 ijms-25-12372-f005:**
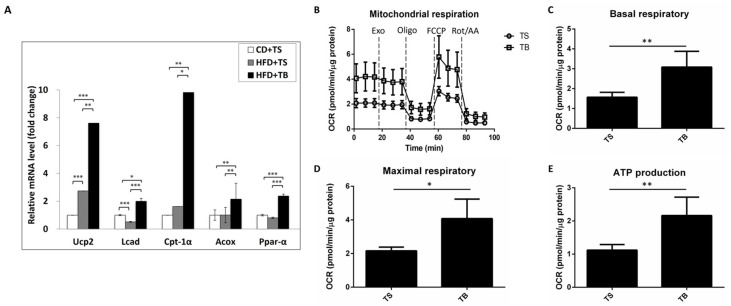
The effects of TB-1 peptide on hepatic fatty acid oxidation. (**A**) Expression levels of fatty acid oxidation-related genes in the liver of the HFD-induced murine MASLD model. (**B**) Oxygen consumption rate (OCR), (**C**) basal respiration, (**D**) maximal respiration, and (**E**) ATP production measured in oleic acid-loaded HepG2 cells following TB-1 peptide treatment. All data are presented as fold changes compared to the expression level of the CD + TS group (*n* = 10 per group). Statistical significance: * *p* < 0.05, ** *p* < 0.01, and *** *p* < 0.001. Abbreviations: CD, chow diet; HFD, high-fat diet; TS, Tat scrambled control peptide; TB, Tat-Beclin-1 peptide; *Ucp2*, uncoupling protein 2; *Lcad*, long-chain acyl-CoA dehydrogenase; *Cpt-1α*, carnitine palmitoyl-transferase 1α; *Acox*, acyl-CoA oxidase; *Ppar-α*, peroxisome proliferator-activated receptor α; OCR, oxygen consumption rate.

**Figure 6 ijms-25-12372-f006:**
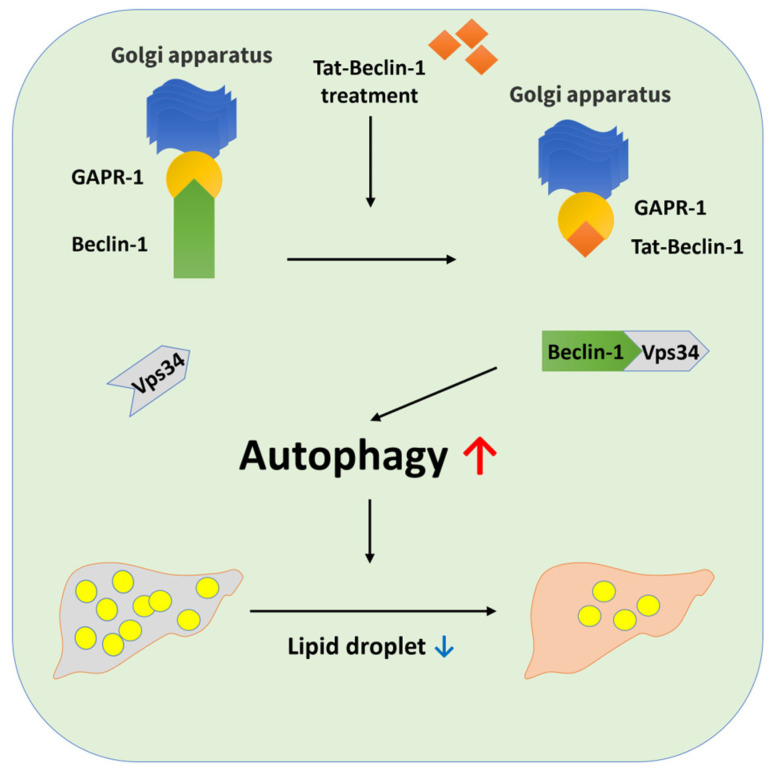
Proposed mechanism of action of TB-1 peptide in promoting autophagy, leading to reduced lipid droplet accumulation. GAPR-1: Golgi-Associated Plant Pathogenesis-Related Protein 1. Red arrow means increase and blue arrow means decrease.

**Table 1 ijms-25-12372-t001:** Serum biochemical parameters of mice treated with TB-1 or scramble peptide.

Variable	CD + TS	HFD + TS	HFD + TB
AST (U/L)	88.6 ± 22.5 ***	226.3 ± 97.2 ^†^	138.7 ± 40.6 ^###^
ALT (U/L)	28.3 ± 9.6 ***	182.7 ± 105.2 ^†††^	96.1 ± 36.2 ^###^
T-CHO (mg/dL)	133.3 ± 12.7 ***	222.0 ± 69.5	206.5 ± 32.2 ^###^
TG (mg/dL)	44.8 ± 22.1	35.5 ± 14.4 ^†††^	65 ± 15.6 ^#^
Glucose (mg/dL)	232.8 ± 67.5	282.4 ± 46.5	245 ± 26.4

All values were expressed as mean ± SD in each group (*n* = 10). Statistical significance: *** *p* < 0.001 (HFD + TS versus CD + TS), ^†^
*p* < 0.05, ^†††^
*p* < 0.001 (HFD + TB-1 control versus HFD + TS control), ^#^
*p* < 0.05, ^###^
*p* < 0.001 (HFD + TB-1 versus CD + TS control). Abbreviations: CD, chow diet; HFD, high fat diet; TS, Tat-Beclin-1 scrambled control peptide; TB, Tat-Beclin-1 peptide; AST, aspartate aminotransferase; ALT, alanine aminotransferase; T-CHO, total cholesterol; TG, triglyceride.

## Data Availability

The original contributions presented in the study are included in the article/Appendix A, further inquiries can be directed to the corresponding author.

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
