# Peer review of "Tat-Beclin-1 Peptide Ameliorates Metabolic Dysfunction-Associated Steatotic Liver Disease by Enhancing Hepatic Autophagy"

_ijms, 2024, doi:10.3390/ijms252212372_

Round 1
Reviewer 1 Report
Comments and Suggestions for Authors
The manuscript can be accepted with minor edits.

Author Response
Comments 1. In Material and methods section, The preparation of TB-1 is not mentioned. Was it injected in mice with saline or any vehicle needs to be mentioned. Add reference for TB-1 dose.
Response 1: Thank you for your question. Both TS and TB-1 peptides are water-soluble, with a solubility of 50 mg/ml in water. We prepared TS and TB-1 peptides in 1X PBS for intraperitoneal injection in mice. The TB-1 peptide was administered at a dose of 20 mg/kg body weight, with supporting references [20-21] included in section 4.4:
- Soria, L. R.; et al., Proc Natl Acad Sci U S A 2018, 115(2), 391-396.
- Tong, M.; et al., Circ Res 2019, 124(9), 1360-1371.
These references are listed in section 4.4 to ensure clarity.
Comments 2. In section 4.9 of materials and method section add primary ab dilution used for WB.
Response 2: Thank you for your valuable suggestion. In response, we have clarified the antibody dilutions used in our Western blotting experiments as follows: LC3B (Cell Signaling, 3868), Beclin-1 (Cell Signaling, 3495), ATG5-12 (Cell Signaling, 4180s), SIRT1 (Proteintech, 13161-1-AP), ULK1 (Cell Signaling, 8054), and phospho-ULK1 (pSer317, Cell Signaling, 12753) were diluted 1,000-fold; SQSTM1/p62 (Abcam, ab109012) was diluted 10,000-fold; and GAPDH was diluted 20,000-fold. We have incorporated these dilution details into section 4.9 for clarity.
Comments 3. In Figure 3A the histology slides can be labelled with lesion for better understanding.
Response 3: Thank you for your suggestion. We have added labels to identify histological lesions in Figures 3A and 3B to enhance clarity. To avoid confusion, black arrow heads are used in Figure 3A to indicate liver steatosis, while black arrows are used in Figure 3B for liver fibrosis.
Comments 4. In the discussion section add a figure explaining the mechanism of action of TB-1 with respect to autophagy in ameliorating MASLD.
Response 4: Thank you for your suggestion. We have added a summary figure in the discussion section to illustrate the mechanism of action of TB-1 peptide in promoting autophagy and its role in reducing lipid droplet accumulation.
Reviewer 2 Report
Comments and Suggestions for Authors
The paper “Tat-Beclin-1 Peptide Ameliorates Metabolic Dysfunction-Associated Steatotic Liver Disease by Enhancing Hepatic Autophagy” is an article evaluating the efficacy of the Tat-Beclin-1 (TB-1) peptide, a specific autophagy inducer, in mitigating MASLD. In the manuscript, Chun-Liang Chen et all. investigate the potential of TB-1 peptide, a potent and specific autophagy inducer, in improving MASLD using a high-fat diet (HFD)-induced murine mode.
In the Introduction, the Authors outlines the importance of autophagy in the development of MASLD and characteristics of Tat-Beclin-1 (TB-1) peptide as a inducer of autophagy processes.
In the Materials and Methods section, the authors describe the material subjected to research and the research methods and tools used. In the experiment Authors cultured HepG2 cells, induced lipid droplet accumulation and autophagy. Lipid droplet quantification was measured by Oil Red O staining. Autophagic vacuoles were stained by CYTO-ID® Autophagy detection kit 2.0. Male C57BL/6JNarl mice were divided into group fed a chow diet and group fed a high-fat diet and injected intraperitoneally with TB-1 peptide. The animals were randomly divided into three groups: CD control (n = 10), HFD+TS (negative control peptide, n = 10), and HFD+TB-1 (n = 10) groups. After 16 weeks of HFD administration, the mice in the treatment group were injected i.p. with TB-1 peptide or TS three times per week for three weeks. Histological analysis of the liver and liver steatosis assessment were conducted using H&E staining. Hepatic fibrosis and the collagen proportionate area (CPA) was assessed using Masson's trichrome (TRI) staining. The serum levels of glucose (GLU), alanine aminotransferase (ALT), alanine transaminase (AST), total cholesterol (T-CHO), and triglycerides (TG) were measured were measured. Blood glucose levels were measured at 0, 15, 30, 45, 60, and 120 minutes after glucose administration. The protein levels of LC3B, SQSTM1/p62, Beclin-1, ATG5-12, SIRT1, ULK1, phospho-ULK1 were determined using Western blotting. Mouse liver RNA extraction and quantitative RT-PCR analysis were performed. The measurement of fatty acid oxidation was conducted.
In the Results section, the Authors present the following:
· TB-1 peptide, but not the TS control peptide, induces autophagy in a dose- and time-dependent manner. Autophagic vacuole staining confirmed these results
· Reduced lipid droplet content in HepG2 cells after TB-1 peptide treatment
· TB-1 peptide induced autophagic vacuoles formation in human hepatocyte
· no significant difference in body weight was observed between the HFD+TB-1 group and the HFD+TS group
· HFD feeding (HFD+TS and HFD+TB-1) resulted in significant increases in serum levels of AST, ALT, and T-CHO. However, treatment with TB-1 peptide significantly improved the elevated levels of AST and ALT. TB-1 peptide treatment significantly improved glucose intolerance induced by the high-fat diet.
· HFD feeding significantly increased hepatic steatosis to 40%. However, TB-1 peptide administration reduced this to 20.3%. TB-1 peptide can attenuate liver fibrosis in MASLD.
· HFD feeding significantly increased the levels of autophagy markers, including LC3-II, Beclin-1, SIRT1, and pULK1/tULK1, while significantly reducing the expression of p62. TB-1 peptide greatly enhances liver autophagy in the context of HFD-induced MASLD.
· TB-1 peptide enhances the expression of fatty acid oxidation-related genes in the context of HFD-induced MASLD, suggesting a potential mechanism for its beneficial effects on lipid metabolism.
· TB-1 peptide activates autophagy and promotes lipid degradation to fuel mitochondrial oxidative phosphorylation.
In the Discussion, based on the results, the Authors conclude that TB-1 peptide is a potential therapeutic agent for MASLD. The Authors discuss their results with available references.
In conclusion the Authors suggest that enhancing autophagy with TB-1 peptide could be a beneficial strategy for managing MASLD by improving lipid metabolism, reducing hepatic steatosis, and alleviating liver fibrosis. Continued research and clinical trials will further elucidate its role and potential in the treatment landscape for MASLD.
The entire manuscript has been enriched with figures presenting histological, molecular, diagnostic and statistical analyses.
The paper entitled “Tat-Beclin-1 Peptide Ameliorates Metabolic Dysfunction-Associated Steatotic Liver Disease by Enhancing Hepatic Autophagy” is clear, comprehensive and has relevance to the field. After reading the manuscript thoroughly, I have no comments to the authors. I believe the manuscript is very good and can be published in present form.
Author Response
Comments 1. The entire manuscript has been enriched with figures presenting histological, molecular, diagnostic and statistical analyses.
The paper entitled “Tat-Beclin-1 Peptide Ameliorates Metabolic Dysfunction-Associated Steatotic Liver Disease by Enhancing Hepatic Autophagy” is clear, comprehensive and has relevance to the field. After reading the manuscript thoroughly, I have no comments to the authors. I believe the manuscript is very good and can be published in present form.
Response 1. Thank you for your valuable feedback and positive assessment of our study.
Reviewer 3 Report
Comments and Suggestions for Authors
In this study, Chun-Liang Chen et. al. reported the effect of Tat-Beclin-1 (TB-1) peptide on ameliorating hepatic steatosis or fibrosis for metabolic dysfunction-associated steatotic liver disease (MASLD) by cellular and animal (mouse) mode. TB-1 enhancing autophagy activity in cellular in HepG2 cell line and MASLD mouse. They suggested TB-1 as a protentional drug for treating MASLD through enhancing autophagy. The authors did a comprehensive study of TB-1 and autophagy effect on hepatic steatosis and fibrosis in cellular and MASLD animal mode. I have some questions:
1. In MASLD animal mode, intraperitoneal injection with TB-1 peptide or TS control peptide was performed for the high fat diet mice from the 16th week to the 19th week. From the weight change (Fig S2), both mice injected with TB-1 or TS control were lost body weights. Were mice lost body weight due to starvation or peritonitis related to repeated intraperitoneal injections? Because starvation may also induce autophagy and body weight loss may decrease hepatic steatosis, the effect of TB-1 on autophagy and steatosis may be added on body weight loss.
2. In this study, MASLD mode mice were feed with high fat diet to 16 weeks. Hepatic steatosis could be found, but the fibrosis change might be rarely seen in only 16 weeks feeding duration. In the manuscript, “ For liver fibrosis evaluation using Masson's trichrome staining, the HFD-induced murine MASLD model showed a liver fibrosis level of 7.3%.” In this study, collagen proportionate area (CPA) measurement was applied. As the authors’ description, the CPA measurement used an “operator-dependent thresholding” to determine the stained area of the section. Was the “7.3%” fibrosis level a mean value or just presented in one experiential animal?
3. In material and methods section, blood biochemical parameters (serum glucose, ALT, AST, total cholesterol, and TG) were measured. Were these data from the blood samples from animals in the16th or the 19th week? Was there any data from the 16th week as a base line for intra- or inter-group analysis?
4. Is there any inflammatory marker or adipokine recorded and available in this study?
5. Minor question: Figure 5, TB-10 is seen. In the text, only TB-1 is found. Dose TB-10 mean “10 μM TB-1”?
Author Response
Comments 1. In MASLD animal mode, intraperitoneal injection with TB-1 peptide or TS control peptide was performed for the high fat diet mice from the 16th week to the 19th week. From the weight change (Fig S2), both mice injected with TB-1 or TS control were lost body weights. Were mice lost body weight due to starvation or peritonitis related to repeated intraperitoneal injections? Because starvation may also induce autophagy and body weight loss may decrease hepatic steatosis, the effect of TB-1 on autophagy and steatosis may be added on body weight loss.
Response 1: Answer: Thank you for your question. During the 3-week course of intraperitoneal injections, mice were fasted only for 6 hours before the OGTT and FITC tests conducted in the final week before sacrifice. The mice displayed no signs of peritonitis, such as reduced appetite, decreased activity, kyphosis, abdominal fluid accumulation, or diarrhea. All three groups (HFD + TB-1, HFD + TS, and CD + TS) that received intraperitoneal injections displayed a similar weight loss trend, regardless of the injected compound. This suggests that the observed weight loss is likely due to the stress associated with repeated intraperitoneal injections rather than the effects of the compounds. Given the similar extent of weight loss across all three groups, the differences observed in liver histology are unlikely to be differentially impacted by this weight change.
Comments 2. In this study, MASLD mode mice were feed with high fat diet to 16 weeks. Hepatic steatosis could be found, but the fibrosis change might be rarely seen in only 16 weeks feeding duration. In the manuscript, “ For liver fibrosis evaluation using Masson's trichrome staining, the HFD-induced murine MASLD model showed a liver fibrosis level of 7.3%.” In this study, collagen proportionate area (CPA) measurement was applied. As the authors’ description, the CPA measurement used an “operator-dependent thresholding” to determine the stained area of the section. Was the “7.3%” fibrosis level a mean value or just presented in one experiential animal?
Response 2: Thank you for your question. The reported 7.3% fibrosis level in our manuscript represents the mean collagen proportionate area (CPA) calculated across all animals in the HFD-induced murine MASLD group. We utilized Masson’s trichrome staining, and the CPA measurement was conducted using an operator-dependent thresholding method to ensure consistent identification of the stained area in each section. To enhance consistency in our fibrosis measurements, this approach was applied consistent criteria across all samples, which minimizes variability and supports uniformity in the fibrosis assessment process.
In this study, the mice were fed a high-fat diet (HFD, 60 kcal% fat + 1.25% cholesterol, customized D12492, Research Diets, Inc.). The addition of cholesterol in the diet contributed to more prominent hepatic fibrosis development within the 16-week feeding duration, making fibrosis changes more noticeable.
[19] Taberner-Cortés A, et al. Treatment with 1.25% cholesterol enriched diet produces severe fatty liver disease characterized by advanced fibrosis and inflammation and impaired autophagy in mice. J Nutr Biochem. 2024 Aug 5;134:109711.
Comments 3. In material and methods section, blood biochemical parameters (serum glucose, ALT, AST, total cholesterol, and TG) were measured. Were these data from the blood samples from animals in the16th or the 19th week? Was there any data from the 16th week as a base line for intra- or inter-group analysis?
Response 3: Thank you for your question. Blood samples were collected exclusively in the 19th week (the final week before sacrifice). For comparison of treatment responses, the CD + TS group was used as the baseline reference.
Comments 4. Is there any inflammatory marker or adipokine recorded and available in this study?
Response 4: Thank you for your question. We measured the expression levels of pro-inflammatory cytokine genes in liver samples. Our results showed an increase in Il-1β and Tnf-α gene expression in the HFD + TS group, with a reduction in the HFD + TB group. This result has been included in Figure S4.
Comments 5. Minor question: Figure 5, TB-10 is seen. In the text, only TB-1 is found. Dose TB-10 mean “10 μM TB-1”?
Response 5: Thank you for pointing that out. We have updated Figure 5 to replace "TB-10" with "TB-1 (10 μM)" to accurately reflect the correct dosage information and avoid any confusion.